# Implementation of a Vaccination Program Based on Epidemic Geospatial Attributes: COVID-19 Pandemic in Ohio as a Case Study and Proof of Concept

**DOI:** 10.3390/vaccines9111242

**Published:** 2021-10-25

**Authors:** Susanne F. Awad, Godfrey Musuka, Zindoga Mukandavire, Dillon Froass, Neil J. MacKinnon, Diego F. Cuadros

**Affiliations:** 1Infectious Disease Epidemiology Group, Weill Cornell Medicine—Qatar, Cornell University, Doha 24144, Qatar; sua2006@qatar-med.cornell.edu; 2World Health Organization Collaborating Centre for Disease Epidemiology Analytics on HIV/AIDS, Sexually Transmitted Infections, and Viral Hepatitis, Weill Cornell Medicine—Qatar, Cornell University, Doha 24144, Qatar; 3Department of Population Health Sciences, Weill Cornell Medicine, Cornell University, New York, NY 10065, USA; 4ICAP, Columbia University, Harare, Zimbabwe; gm2660@cumc.columbia.edu; 5Centre for Data Science and Artificial Intelligence, Emirates Aviation University, Dubai 53044, United Arab Emirates; zindoga.mukandavire@emirates.com; 6College of Medicine, University of Cincinnati, Cincinnati, OH 45221, USA; froassdc@mail.uc.edu; 7Department of Population Health Sciences, Medical College of Georgia, Augusta University, Augusta, GA 30912, USA; nmackinnon@augusta.edu; 8Department of Geography and Geographic Information Science, University of Cincinnati, Cincinnati, OH 45221, USA; 9Health Geography and Disease Modeling Laboratory, University of Cincinnati, Cincinnati, OH 45221, USA

**Keywords:** vaccination program, geospatial attributes, spatial epidemiology, disease mapping, COVID-19, mathematical model

## Abstract

Geospatial vaccine uptake is a critical factor in designing strategies that maximize the population-level impact of a vaccination program. This study uses an innovative spatiotemporal model to assess the impact of vaccination distribution strategies based on disease geospatial attributes and population-level risk assessment. For proof of concept, we adapted a spatially explicit COVID-19 model to investigate a hypothetical geospatial targeting of COVID-19 vaccine rollout in Ohio, United States, at the early phase of COVID-19 pandemic. The population-level deterministic compartmental model, incorporating spatial-geographic components at the county level, was formulated using a set of differential equations stratifying the population according to vaccination status and disease epidemiological characteristics. Three different hypothetical scenarios focusing on geographical subpopulation targeting (areas with high versus low infection intensity) were investigated. Our results suggest that a vaccine program that distributes vaccines equally across the entire state effectively averts infections and hospitalizations (2954 and 165 cases, respectively). However, in a context with equitable vaccine allocation, the number of COVID-19 cases in high infection intensity areas will remain high; the cumulative number of cases remained >30,000 cases. A vaccine program that initially targets high infection intensity areas has the most significant impact in reducing new COVID-19 cases and infection-related hospitalizations (3756 and 213 infections, respectively). Our approach demonstrates the importance of factoring geospatial attributes to the design and implementation of vaccination programs in a context with limited resources during the early stage of the vaccine rollout.

## 1. Introduction

The rollout of reactive and preemptive vaccination programs to prevent the spread of diseases and lower mortality rates has been of utmost importance to end epidemics and pandemics [1]. Insights from vaccine campaigns against smallpox, polio, cholera, and H1N1, among others, have offered great lessons relevant to the rollout of future vaccines [2]. As countries have embarked on a massive implementation and rollout of Coronavirus Disease 2019 (COVID-19) vaccines (and of future vaccination programs), understanding the challenges faced in previous and current vaccines campaigns is critical in designing strategies that maximize the population-level benefits.

The most meaningful and challenging policy is to design an effective vaccine program that achieves optimal impact with highest population-level benefits. Vaccine hesitancy, limited availability, staff shortages, budgets, and supply constraints can plague the distribution process of vaccines, and thus a campaign’s overall impact, particularly during the early stage of the vaccine rollout. For instance, in countries like the United States (US) and Japan, vaccine hesitancy has been a primary cause of low uptake for the vaccine against human papillomavirus (HPV) [3,4], resulting in deaths that otherwise could have been prevented [4].

A challenging question related to early vaccine implementation is *“Who should get vaccinated first?”* Besides demographic prioritization [5,6,7,8,9,10], evidence suggests that infectious diseases, including respiratory infections, have substantial geographical variations in intensity and transmission range that are induced by an uneven distribution of vulnerable populations and risk factors [11,12,13,14,15]. These variations facilitate (or hamper) the spatial diffusion of the pathogen [11,12,13,14,15]. For instance, regions across the US are unequally affected by the COVID-19 pandemic with several socioeconomic, health-related (e.g., comorbidities), and other geospatial factors being critical drivers of the geographical disparities of COVID-19 related hospitalizations and deaths [16,17,18,19,20]. Moreover, vulnerable populations such as ethnic minorities, socially disadvantaged poor populations, and hard-to-reach rural communities could suffer lower vaccination coverage as a result of the practical difficulties in implementing an immunization program [21]. Therefore, information on these geospatial disparities of a pandemic and the distribution of vulnerable populations can become a crucial component for designing and implementing a successful vaccination campaign.

The successful application of geospatial analysis in the identification of clusters for cholera [10] in identifying “hot” and “cold” spots of diseases, the monitoring of risk behaviors of recently circumcised men for HIV prevention efforts [6,22,23,24,25], among other applications, have illustrated the effective mechanism of geospatial analysis for supporting decisions regarding pandemic prediction and hospital capacity management during outbreaks. However, geospatial analysis in vaccine distribution and prioritization at different spatial resolutions is poorly utilized, allowing further investigation of such novel topic.

Against this background, we aim to determine the best approach to early vaccine distribution on a spatial scale. Specifically, the purpose of this study is to develop a tool that determines which type of vaccine distribution strategy would be most effective in minimizing both hospitalizations and new cases during the implementation of a vaccination campaign, when vaccine availability is limited and the need for identifying the most vulnerable populations is high. In order to explore this concept, we assess and evaluate the impact of different geospatial rollout strategies for COVID-19 vaccine during the early stage of vaccine rollout in Ohio, US.

We use COVID-19 as an example, as COVID-19 vaccine campaigns are just the latest chapter in a long history of vaccination rollout, and there is a wealth of geospatial data available that has been collected to track the progression of the pandemic [21]. We evaluate the developed tool in Ohio because the spread and intensity of the infection varied substantially across the different spatial risk areas with high and low infection intensity across the state. Furthermore, there is detailed geographical data [26] and a published spatial population-level COVID-19 model for Ohio [20,26] that can be adapted for the purpose of this study.

## 2. Methods

### 2.1. Model Structure

A spatial population-level deterministic mathematical model was developed to simulate the transmission dynamics and spread of COVID-19 in Ohio and to investigate the potential impact of the novel COVID-19 vaccines. The basic model structure and formulation details have been previously described in detail [20,26]. Briefly, the spatially explicit model, incorporating geographic connectivity information at the county level, was modeled using a set of differential equations stratifying the population according to susceptibility (S; defined as an individual free of infection, but who can acquire the infection from an infected individual), confirmed infections (I), hospitalized (H), and intensive care unit (ICU) admitted, recovered (R) and deceased (D). The Susceptible-Infected-Hospitalized-Recovered-Dead (SIHRD) COVID-19 model was classified into four different spatial risk groups based on the defined connectivity index for each county. Based on earlier studies [20,26], *spatial risk group 1* was defined as counties with airports with more than 50,000 passengers per year, *spatial risk group 2* was defined as counties adjacent to the counties with airports, *spatial risk group 3* was defined counties with main highways crossing the county, and *spatial risk group 4* was defined as low connected counties not adjacent to counties with airports or being crossed by main highways. The model incorporated the impact of public health interventions such as social distancing and “stay home orders” implemented in Ohio.

In this study, the compartmental population model was further stratified to include vaccine status. As noted earlier, the model uses the COVID-19 pandemic as a proof of concept, but it does not intend to accurately project the actual impact of current available vaccines. Therefore, we assumed that a single dose vaccine (i.e., Janssen vaccine [27]) was incorporated into the model through a distinct and separate COVID-19 compartment capturing the number of vaccinated individuals. Equations and details of the extended model structure are given in Appendix A.

### 2.2. Data Sources and Model Fitting

The model was parameterized using empirical evidence for COVID-19 and was fitted to data obtained from the Ohio Department of Health [28] using MATLAB R2019a [29]. The data used for model fitting were cumulative number of confirmed COVID-19 cases, COVID-19-related hospitalizations, ICU admissions, and COVID-19 deaths in each county in Ohio from 1 March 2020 to 21 October 2020. This time interval captures the epidemic dynamics in Ohio during the implementations of non-pharmaceutical interventions early April, and the easing of these interventions at the beginning of June 2020. Data from Marion and Pickaway counties, which experienced unusual infection outbreaks in prisons, were excluded in order to simulate COVID-19 transmission dynamics in the general population [26]. For each spatial risk group, the hazard rate, infection rate which flows from other spatial risk groups, hospitalization rate, hospitals discharge rate, ICU admission rate, recovery rate from ICU admission, and death rate were estimated by fitting the model to data. A nonlinear least-squares data fitting method, based on the Nelder–Mead simplex algorithm [30], was used to minimize the sum of squares between data points and model predictions.

### 2.3. Vaccine Scale-Up Scenarios

For all modeled vaccine scenarios, we assumed that the early implementation of the vaccine rollout with limited vaccines was initiated 300 days into the COVID-19 epidemic (i.e., 1 January 2021) and scaled up at a fixed rate for 30 days. We estimated the effect of the vaccination campaign during the following four months. As vaccines might not be available as a mass vaccine at the beginning of a vaccine program, for this modeling proof of concept exercise, we assumed that limited number of vaccines were available; one-million vaccine doses would be allocated and distributed across Ohio within 30 days. We also implicitly accounted for the effect of response rates and willingness to be vaccinated through the lower vaccination coverage.

The efficacy of vaccines against COVID-19 acquisition was incorporated in the model assuming it generates proportional reduction in the risk of COVID-19 acquisition among vaccinated individuals. The vaccine efficacy in reducing susceptibility was assumed to be 90%, on average, based estimations from the novel vaccines currently available. Given that the impact of the COVID-19 vaccine is assessed here as a proof-of-concept during the early stage of the pandemic, we did not incorporate the variation in transmissibility arising from emerging COVID-19 virus variants. The effect of the vaccine was assumed not to wane within the simulation (i.e., at least one year of protection against COVID-19 once vaccinated). The immunity among individuals recovered from infection was also assumed to last at least one year.

### 2.4. Measures of Impact

The primary outcome of interest from the model was the number of averted cases. This was calculated by comparing the number of new cases in the vaccine scenario versus the baseline no-vaccine scenario. The number of COVID-19 related hospitalizations that the vaccine can avert was also calculated by comparing the vaccine and no-vaccine scenario. The impact of the vaccine program scenarios was assessed through the effectiveness of the vaccine program—the number of vaccinations required to avert one infection (or one COVID-19 related hospitalization) in the population. Effectiveness was estimated by dividing the number of COVID-19 vaccine doses needed, by the number of COVID-19 cases, over a time horizon.

### 2.5. Analysis Plan

We evaluated the vaccine prioritization for subpopulations based on geographical risk areas. We evaluated three different scenarios focusing on subpopulation targeting program based on the four spatial risk groups previously described (Group 1: counties with airports with more than 50,000 passengers per year, Group 2: counties surrounding the counties with airports, Group 3: counties with main highways crossing the county, and Group 4: counties not adjacent to counties with airports or being crossed by main highways-low risk areas).

In the first vaccine program scenario, one-million vaccine doses were distributed homogeneously across Ohio’s different spatial risk areas (i.e., 250,000 doses allocated to each risk area). In the second vaccine program scenario, 60% of all vaccines were administered to *Groups 3* and *4* (i.e., identified as areas with low infection intensity), while 40% of the vaccines were equally distributed across *Group 1* and *Group 2* (i.e., identified as areas with high infection intensity). In the third vaccine program scenario, 90% were equally distributed across *Groups 1* and *2,* and only 10% of all vaccines were allocated to *Groups 3* and *4*. Given that countries could make vaccine allocation decisions based on easy-to-reach rather than on infection intensity, the impact of such choice is presented in scenario 2, and thus, we wanted to compare the quantitative impact of such choice.

### 2.6. Uncertainty and Sensitivity Analyses

A multivariable uncertainty analysis was conducted to determine the range of uncertainty around model predictions for the effectiveness of the different vaccine scenarios. We drew 10,000 samples using a Latin Hypercube sampling from a multidimensional distribution of the model parameters in MATLAB R2019a [29]. Parameter values were selected from ranges specified by assuming ±30% uncertainty around parameters’ point estimates. The resulting distributions of estimates across all runs were used to calculate the 95% credible intervals (Crls) for model predictions [31].

Vaccine efficacy was assessed through a sensitivity analysis. Consequently, the three scenarios reported above were conducted assuming a 90% vaccine efficacy for vaccines distributed to high transmission intensity areas (*Groups 1* and *2*) and a 50% vaccine efficacy for vaccines distributed to low transmission intensity areas (i.e., *Groups 3* and *4*).

## 3. Results

The model estimated a total number of 230,107 (95% Crl 176,725–323,118) COVID-19 cases compared to the 180,130 confirmed cases (excluding Marion and Pickaway counties) reported by 18 October 2020 in Ohio (Appendix A). In the no-vaccination scenario, between 1 March 2020 and 1 May 2021, the predicted cumulative number of new COVID-19 cases (divided by the population size in each risk area) was 12,818 (CrI 12,261–13,527) per 100,000 people in Group 1, 10,833 (CrI 9296–12,314) per 100,000 people in Group 2, 9399 (CrI 7697–10,905) per 100,000 people in Group 3, and 8908 (CrI 7304–10,479) per 100,000 people in Group 4 (Figure 1). Similarly, the predicted cumulative number of hospitalizations due to COVID-19 (divided by the population size in each risk area) was 913 (CrI 899-923) per 100,000 people in Group 1, 615 (CrI 527–701) per 100,000 people in Group 2, 350 (CrI 287–407) per 100,000 people in Group 3, and 243 (CrI 199–286) per 100,000 people in Group 4 (Appendix A).

### 3.1. Impact of Various Vaccine Campaigns on COVID-19 Infections

By introducing vaccination on 1 January 2021, the total number of COVID-19 infections averted in Ohio, by 1 May 2021, ranged between 2149 cases (CrI 1997–2177; *scenario 2*) and 3756 cases (CrI 3528–3786; *scenario 3*; Figure 2A) per 100,000 people. The total number of vaccinations needed to avert one infection ranged between 3.15 (CrI 3.1–3.41; *scenario 3*) and 5.17 (CrI 4.18–5.49; *scenario 2;*
Figure 2A and Appendix A).

Specifically, the cumulative number of new COVID-19 infections in Group 1 ranged between 8588 and 10,599 per 100,000 people showing a reduction in incidence ranging between 17.3–33.0% (Figure 1A). The most substantial reduction was achieved by the vaccination scenario in which 10% of all vaccines were allocated to Groups 3 and 4 (low transmission intensity areas), while 90% of the vaccines were administered to Groups 1 and 2 (high transmission intensity areas; *scenario 3*; Figure 1A). The least reduction was achieved by the vaccination scenario in which 60% of all vaccines were allocated to Groups 3 and 4, while 40% of the vaccines were allocated to Groups 1 and 2 (*scenario 2*; Figure 1A). Similarly, the cumulative number of new COVID-19 infections in Group 2 ranged between 7019 and 8884 per 100,000 populations (an 18.0–35.2% reduction in the number of cases; Figure 1B) and in Group 3 ranged between 7019 and 8884 per 100,000 populations (a 20.5–35.2% reduction in the number of cases; Figure 1C). In Group 4, the cumulative number of new COVID-19 infections were 6000 per 100,000 populations (approximately 32% reduction in incidence) regardless of vaccination scenario (Figure 1D).

### 3.2. Impact of Various Vaccine Campaigns on COVID-19-Related Hospitalizations

The estimated total number of COVID-19 hospitalizations averted in Ohio, by 1 May 2021, ranged between 116 (CrI 100–117; *scenario 2*) and 213 (CrI 193–214; *scenario 3*; Figure 2B) per 100,000 people. The total number of vaccinations needed to avert one hospitalization ranged between 56 (CrI 55–52; *scenario 3*) and 96 (CrI 94–103; *scenario 2*; Figure 2B, and Appendix A).

Specifically, the cumulative number of COVID-19 related hospitalizations in Group 1 ranged between 614 and 757 per 100,000 people meaning a reduction in incidence ranging between 17.1–32.7% (Appendix A). The most significant decrease in hospitalizations was achieved by vaccination *scenario 3*, while the smallest reduction was achieved by vaccination *scenario 2*. Similarly, the cumulative number of COVID-19 related hospitalizations in Group 2 ranged between 400 and 506 per 100,000 people (a 17.7–34.9% reduction in the number of hospitalizations (Appendix A), and Group 3 it ranged between 233 and 279 per 100,000 people (a 20.2–33.5% reduction in the number of hospitalizations; (Appendix A). In Group 4, the estimated cumulative number of COVID-19-related hospitalizations 160 per 100,000 people (around 32% reduction in hospitalizations) regardless of vaccination scenario (Appendix A).

### 3.3. Spatiotemporal Dynamics of the Epidemic

Without vaccination, the spread and intensity of the infection varied substantially across the different spatial risk areas (Figure 3). The epidemic was spreading faster and affecting more individuals in counties with local air hubs from Group 1. In contrast, the less connected counties from Group 4 experienced a slower and less severe spread of the disease (Figure 3 maps on the top). In the scenario in which approximately one-million vaccines were distributed equally across Ohio’s different spatial risk areas (vaccination *scenario 1*), the spread of the infections became lower across the counties. However, the cumulative number of cases in Group 1 remained >30,000 cases, while the cumulative number of cases in Group 2 and Group 3 were reduced from 5001–10,000 and 1501–5000 cases, respectively, to less than 1500 cases. In *scenario 3*, in which 10% of all vaccines were allocated to Group 3 and Group 4, while 90% of the vaccines were equally distributed across Group 1 and Group 2, the spread of the infection became more homogenous across the different spatial risk groups. Still, Group 1 experienced between 10,000 and 30,000 cases by 1 May 2021 (Figure 3).

Similarly, for the cumulative number of hospitalizations, substantial spatial disparities in the number of hospitalizations among counties in Ohio were observed in the no-intervention scenario. Most areas were predicted to have more than 100 admissions in the observed period (Figure 3 maps on the second row). Once vaccination was introduced (regardless of vaccination scenario), hospitalizations were reduced across the different spatial risk areas. In *scenario 3*, there were still disparities in the number of hospitalizations among the different counties; however, for most counties, less than 100 hospitalizations were predicted in the observed period.

In the sensitivity analysis in which different vaccine efficacies were assumed for the rural areas (Groups 3 and 4; efficacy of 50%) compared to urban areas (Groups 1 and 2; efficacy of 90%), population-level impacts of the vaccination scenarios were similar to the original analyses (Appendix A).

## 4. Discussion

This study assesses the application and effectiveness of designing geospatial targeting vaccine implementation programs. A geospatial subpopulation prioritization tool, such as the one we have described here, is not meant to exclude anyone from a vaccine program (or discourage equitable vaccine distribution) but instead demonstrates the implications of intensifying demand, creation, and service availability for specific subpopulations in the early stage of vaccination rollout, or when vaccine availability is an important limitation [33]. As a proof-of-concept, we investigated the geospatial targeting of the most recent COVID-19 vaccine rollout on the early phase for COVID-19 outbreak in Ohio. This approach might be used to evaluate other intervention strategies or regions, given that a robust model and geospatial data are available to describe the natural history and transmission dynamics of a disease. Of note, this study focused on the implication of geospatial vaccination coverage and did not consider the impact of vaccinations based on other factors like ethnicity, age, etc.

Our results suggest that a vaccine program that distributes vaccines equally across areas with high and low infection intensity effectively averts infections and hospitalizations (Figure 2). However, the number of cases in areas with high infection intensity experienced in highly connected areas will remain high. On the other hand, a vaccine program that initially targets regions with high infection intensity has at the population-level the greatest impact in reducing the number of new COVID-19 cases and infection-related hospitalizations during the early stage of the vaccine rollout. In contrast, our model suggests that a program that initially targets areas with low infection intensity characterized by the low connectivity in these areas is the least effective at achieving this goal. These findings highlight that for settings with limited vaccines, focusing the early phases of vaccine distribution on areas with high infection intensity informed by specific geospatial attributes, in this case the high connectivity of these areas, could be an optimal strategy to initially reduce transmission in the entire population.

However, note that epidemic dynamics might change over time, and areas of high transmission intensity can vary over time, fueled by the emergence of new geospatial factors driving the epidemic at different phases. As for Ohio and the other states in the US, and other countries globally, the early rollout of the COVID-19 vaccine was initially based on demographic characteristics rather than geospatial attributes and the location of subpopulations at risk [21,33]. The uptake of COVID-19 vaccines in early 2021 was rapid (not just in Ohio, but across US), but following this initial rapid uptake, the vaccination rate slowed down considerably due to several challenges such as vaccine hesitancy, reaching vulnerable populations, socially disadvantaged poor populations, and hard to reach rural communities. Therefore, for a successful vaccination strategy, the geospatial attributes and the location of subpopulations at risk should be continuously revisited.

Communities with low vaccination rates could become infection reservoirs from where infections could circulate, facilitated by the movement of people between low-risk communities and the rest of the population. These communities with low vaccination rates could compromise the targeted herd immunity by creating pockets of infections that sustain outbreaks of transmission in the entire region. In particular for COVID-19, this situation could linger for years as areas with low infection intensity and low vaccine coverage could create disease reservoirs that could generate a chain of infections in the future. In fact, COVID-19 vaccination and transmission in the US in the mid-2021 is an example of such threat. Areas that were previously identified as low transmission areas, which also have low vaccine uptake, were experiencing a high COVID-19 transmission wave by mid-2021 and have become the driver of the epidemic hubs in the country. Once the protective mechanism of COVID-19 wanes, these hubs could cause new epidemics not only in low-coverage areas, but throughout the whole country.

### Limitations of the Study

This study has limitations worth noting. The connectivity index for each county was based only on one geospatial attribute: “connectivity enhanced by airports and main roads”; yet transmission dynamics and disease trajectories could potentially be influenced by other geospatial and population characteristics. However, including more geospatial features might substantially increase model complexity without significantly affecting the conclusions of the study. Moreover, data from Marion and Pickaway counties in Ohio, which experienced explosive confined COVID-19 outbreaks occurring in different prisons, were not included in the analysis as the scope was to assess the impact of vaccination scenarios in the general population. However, confined outbreaks such as those could become an essential element in a vaccine campaign and the spread of the disease as they could create infection reservoirs around a specific setting. Furthermore, the model was fitted using COVID-19 confirmed cases, which might be substantially underestimated or vary depending on changes in testing intensity over time. However, our model utilized available data for confirmed cases, hospitalizations, and deaths that simultaneously affected the tracking the disease dynamics in the model and potentially accurately inform patterns of disease transmission in space and over time. Additionally, our modeling main goal was to simulate general spatio-temporal disease dynamics based on patterns shown by available data rather than generate accurate predictions. Likewise, the SIHRD COVID-19 model and parameters utilized in this study were based on fitting the model to data from only Ohio; globally and within countries, the COVID-19 pandemic comprises a series of epidemics of different intensities, each of which had implemented different control measures at different times. This might generate substantial differences in the dynamics of the disease and the impact of geospatial vaccination distribution. For this reason, the generalization of the conclusions for this study should be interpreted with caution. It would be essential to assess the impact of geospatial vaccine distribution using data from other locations or regions. In the model, we did not consider elaborated features of COVID-19 such as higher transmissibility due to the emergence of virus variants, that might otherwise affect the predictions of our model. Further studies to assess the impact of the emergence of virus variants on vaccinations strategies are guaranteed. Additionally, like in any epidemic, strong political will and leadership will be required to mobilize the necessary statewide, regional, and local consensus, financial and other resources to implement this innovative approach to epidemic response.

Despite these limitations, this study presents the first spatiotemporal model to assess the impact and effectiveness of vaccination distribution strategies based on the spatial dynamics of the disease targeting geographical areas where vulnerable populations at high risk of the infection reside.

## 5. Conclusions

By using COVID-19 in Ohio as a case study, we found that inclusion of geospatial attributes can be an important consideration for the design and implementation of a successful vaccination rollout when a limited number of vaccines are available and need to be distributed to maximize the population-based impact of the vaccine campaign.

This study presents a modeling approach for predicting the most effective population-level vaccination distribution strategy based on disease geospatial attributes (e.g., geographic connectivity information) and population-level risk assessment (e.g., susceptibility, confirmed infections, etc.). In cases of limited vaccines, early phases of vaccine distribution should focus on areas with high infection intensity informed by geospatial attributes. Such an approach could be a cornerstone of effective vaccine microplanning. While traditional microplanning relies on census data to define target populations based on demographic attributes, geographical information systems (GIS) microplanning can also identify challenges and reach unreached people. GIS microplanning has, for example, been used very effectively for a polio vaccination program and assisted in eliminating polio in Nigeria [34]. Further, this innovative geospatial approach should not be limited to vaccination per se but has applicability in contact tracing, mapping variants using GIS to assist the clinical response. Additionally, it will help us identify clusters of individuals refusing vaccinations so that a targeted response can be implemented. We can also be able to map areas where vaccine breakthroughs have occurred.

## Figures and Tables

**Figure 1 vaccines-09-01242-f001:**
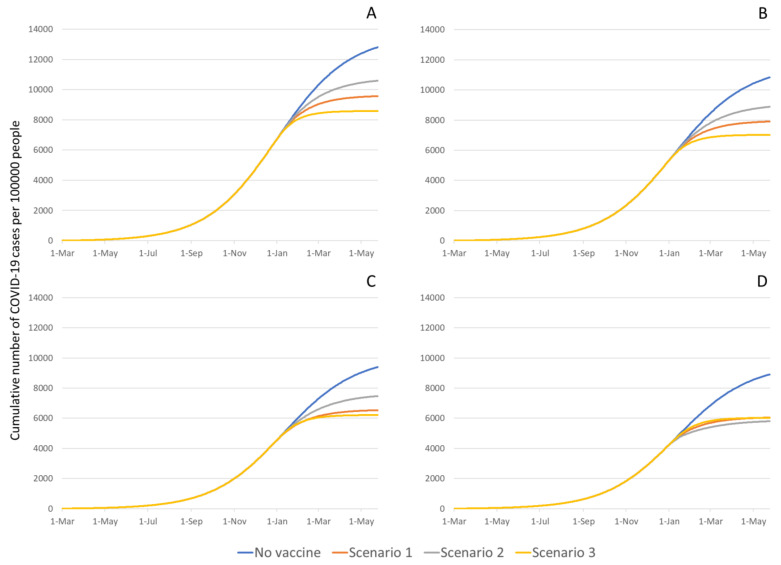
COVID-19 cumulative cases per 100,000 people in each risk area group. (**A**) Group 1-counties with airports with more than 50,000 passengers per year, (**B**) Group 2-counties surrounding the counties with airports, (**C**) Group 3-counties with main highways crossing the county, and (**D**) Group 4-counties with main highways crossing the county. The cumulative number of cases was defined as the number of cases in each risk area divided by the population size in each risk area multiplied by 100,000 people. The no-vaccine prediction was based on best fitted model to data obtained from the Ohio Department of Health [28].

**Figure 2 vaccines-09-01242-f002:**
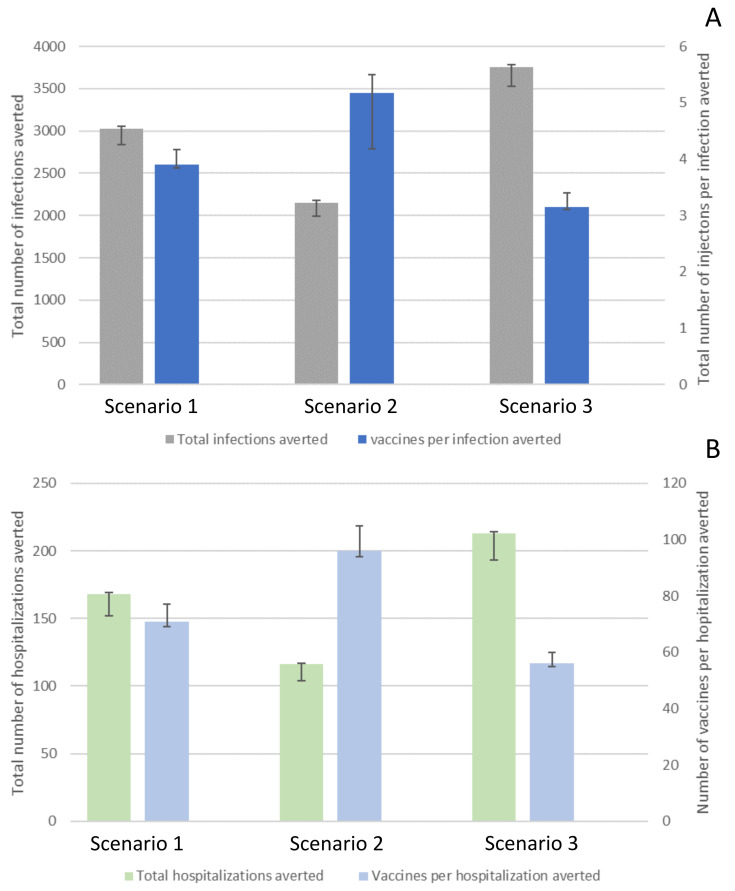
Impact of different vaccination scenarios on COVID-19-related infections (Panel **A**) and hospitalizations (Panel **B**). The effect was defined in averted cases, and the number of vaccinations needed per case/hospitalization averted (i.e., population-level effectiveness of a vaccination program). Scenario 1, one-million vaccine doses were distributed homogeneously across Ohio’s different spatial risk areas. Scenario 2, 60% of all vaccines were administered to Group 3 and Group 4 (i.e., identified as areas with low infection intensity), while 40% of the vaccines were equally distributed across Group 1 and Group 2 (i.e., identified as areas with high infection intensity). Scenario 3, 90% were equally distributed across Group 1 and Group 2, and only 10% of all vaccines were allocated to Group 3 and Group 4.

**Figure 3 vaccines-09-01242-f003:**
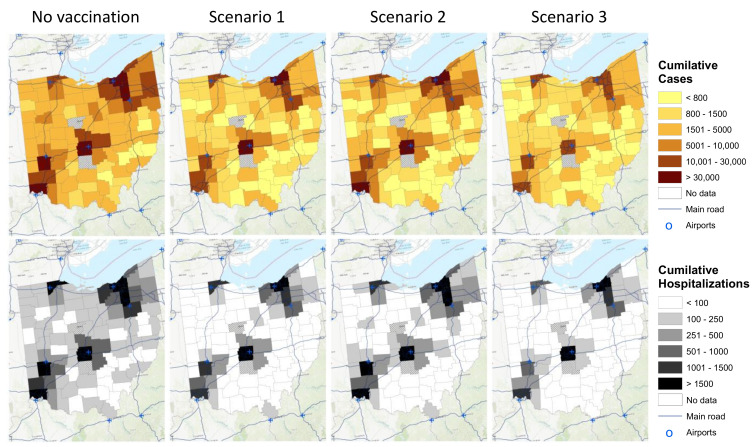
The adjusted spatial model estimated the spatiotemporal dynamics of the cumulative number of COVID-19 cases (maps on the top) and the cumulative number of COVID-19 hospitalizations (maps on the bottom). Maps were created using ArcGIS^®^ by ESRI version 10.3 (http://www.esri.com, accessed on 10 August 2021) [32]. Scenario 1, one-million vaccine doses were distributed homogeneously across Ohio’s different spatial risk areas. Scenario 2, 60% of all vaccines were administered to Group 3 and Group 4 (i.e., identified as areas with low infection intensity), while 40% of the vaccines were equally distributed across Group 1 and Group 2 (i.e., identified as areas with high infection intensity). Scenario 3, 90% were equally distributed across Group 1 and Group 2, and only 10% of all vaccines were allocated to Group 3 and Group 4.

## Data Availability

Data are available in a public, open-access repository. The data that support the findings of this study are available from the Ohio Department of Health (https://coronavirus.ohio.gov/wps/portal/gov/covid-19/home accessed on 8 August 2021).

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
