# Peer review of "Implementation of a Vaccination Program Based on Epidemic Geospatial Attributes: COVID-19 Pandemic in Ohio as a Case Study and Proof of Concept"

_vaccines, 2021, doi:10.3390/vaccines9111242_

Round 1

Reviewer 1 Report

Hypothetical scenarios have been used to determine the best approach, based on geospatial coverage to distribute vaccines to  avert infections and hospitalisations. Data relating from Covid 19 cases in Ohio was used in a mathematical model but his approach will be applicable for other infections. The method has been used in previous studies. The manuscript is generally well written and conclusions valid but could be better organised.

The figure legends should come below each figure. 

In the results section the number of cases given at the bottom of pages 4 and 5 is difficult to follow. This would clearer in well headed tables.

Author Response

Reviewer #1:
Hypothetical scenarios have been used to determine the best approach, based on geospatial
coverage to distribute vaccines to avert infections and hospitalisations. Data relating from
Covid 19 cases in Ohio was used in a mathematical model but his approach will be applicable
for other infections. The method has been used in previous studies. The manuscript is generally
well written and conclusions valid but could be better organised.

Comment: We thank the reviewer for assessing our work and for the constructive feedback on
our manuscript that enriched the article and improved its readability. Please find below a point-
by-point reply addressing the reviewer’s comments.

1. The figure legends should come below each figure.

Answer: Unfortunately, the placement of the figures and figure titles are as per journal
requirement and cannot be placed differently. However, to address the reviewer’s suggestion we
have now in the revised Results changed the formatting of Figure 2 and 3 to match that of Figure
1 (i.e., not merged with text) thus, clearly distinguishing the figure title from main text (Figure 2
and Figure 3).

2. In the results section the number of cases given at the bottom of pages 4 and 5 is difficult to
follow. This would clearer in well headed tables.

Answer: We have included a table summarizing these results in Supplementary Materials (Table
S3)

Reviewer 2 Report

This manuscript by Awad et al. presents a modelling study for predicting the most effective population-level vaccination strategy based on disease geospatial attributes (e.g., geographic connectivity information) and population-level risk assessment (e.g., susceptibility, confirmed infections, etc.). The authors used the COVID-19 pandemic and the vaccine rollout as a proof-of-concept test case for an adapted spatiotemporal model to assess vaccination distribution strategies based on disease geospatial attributes. I found this study to be quite interesting, and the conclusion informative (i.e., that in cases of limited vaccines, early phases of vaccine distribution should focus on areas with high infection intensity informed by geospatial attributes). Overall, the manuscript is well-written, and the results are clearly presented. My specific comments are given below.

  1. Why was the State of Ohio selected as the proof-of-concept test case? Authors should provide a rationale for their selection of Ohio in the Methods.
  2. In the Methods, the authors have “Measures of impact” at line 155. It is unclear whether this is meant to be a subheading for a separate subsection in Methods since it is indented but not italicized. Please clarify.
  3. For all the figures in this manuscript (Figs. 1, 2, and 3), please place the figure title and legend below the corresponding figure and sent it apart from the text. The legends for Figures 2 and 3, in particular, merge with the text.
  4. Please remove “is” in line 412.

Author Response

Reviewer #2:

This manuscript by Awad et al. presents a modelling study for predicting the most effective
population-level vaccination strategy based on disease geospatial attributes (e.g., geographic
connectivity information) and population-level risk assessment (e.g., susceptibility, confirmed
infections, etc.). The authors used the COVID-19 pandemic and the vaccine rollout as a proof-
of-concept test case for an adapted spatiotemporal model to assess vaccination distribution
strategies based on disease geospatial attributes. I found this study to be quite interesting, and
the conclusion informative (i.e., that in cases of limited vaccines, early phases of vaccine
distribution should focus on areas with high infection intensity informed by geospatial
attributes). Overall, the manuscript is well-written, and the results are clearly presented. My
specific comments are given below.

Comment: We thank the reviewer for assessing our work and for the constructive feedback on
our manuscript that enriched the article and improved its readability. Please find below a point-
by-point reply addressing the reviewer’s comments.

  1. Why was the State of Ohio selected as the proof-of-concept test case? Authors should provide
    a rationale for their selection of Ohio in the Methods.

    Answer: We thank the reviewer for the comment. We have now added the rationale behind the
    choice in the revised Introduction (lines 106-110 in the Introduction section of the marked
    document).
  2. In the Methods, the authors have “Measures of impact” at line 155. It is unclear whether this
    is meant to be a subheading for a separate subsection in Methods since it is indented but not
    italicized. Please clarify.

    Answer: We apologize for the confusion. “Measures of impact is indeed a subheading. The
    subtitle has now been italicized to in the revised Methods (line 174 of Methods in he marked
    documetn).
  3. For all the figures in this manuscript (Figs. 1, 2, and 3), please place the figure title and
    legend below the corresponding figure and sent it apart from the text. The legends for
    Figures 2 and 3, in particular, merge with the text.

    Answer: Unfortunately, the placement of the figures and figure titles are as per journal
    requirement and cannot be placed differently. However, to address the reviewer’s suggestion we
    have now in the revised Results changed the formatting of Figure 2 and 3 to match that of Figure
    1 (i.e., not merged with text) thus, clearly distinguishing the figure title from main text (Figure 2
    and Figure 3).
  4. Please remove “is” in line 412
    Answer: We thank the reviewer for pointing this out. This has now been removed in the revised
    Discussion.

Reviewer 3 Report

In the manuscript "Implementation of a vaccination program based on epidemic geospatial attributes: COVID-19 pandemic in Ohio as a case study and proof of concept” Susanne Awad and colleagues determined which vaccine distribution strategy - on a spatial scale - would be the most effective in minimizing both hospitalizations and new COVID-19 cases in Ohio (US) in particular in a context characterized by limited resources (i.e., availability) during the early stage of the vaccine rollout.

They use an innovative spatiotemporal model to assess the impact of vaccination distribution strategies based on disease geospatial attributes and population-level risk assessment, and demonstrated that a vaccine program that initially targets high infection intensity areas – compared to an equitable vaccination - has the most significant impact in reducing new cases and infection-related hospitalizations.

Although the math under the study is beyond my competences I found very interesting and useful this approach in particular in the context of a pandemic; however, some inaccuracies and gaps particularly in the virologic facets are present and deserve attention.

Here some comments/concerns (among others).

The authors stated “this approach can be generalized to any other intervention strategy or any region” therefore it could be very interesting and convincing if they show this approach is valid in a different region (nation, why not) and also for other well-known diseases (i.e. influenza, measles).

Authors should consider the transmissibility variation in the context of the circulating variants that represents a very important factor and make differences in the spreading of the infection.

In Figure S1 (and also in the formula), authors should consider that deaths may occur also out of ICU. Moreover, after dismission people may be susceptible and/or receive vaccination. (Note: in the contest of high-transmissible variant, vaccination do not grant a 100% protection from infection this make the difference as variable).

Authors may define and explain what they refer to as susceptible.

Not sure about what is the bottom-line rationale of the Scenario 2.

Authors should explain which is the weight of the containment measures, actually this is a known and wide used index defined as stringency that they should include in the equation.

In Line 367-369 authors stated, without reference, that “… for a successful vaccination strategy, vaccines should be distributed equally in all areas and covering all population groups”, not sure if it is sharp to state that in the context of a study where authors’ message is different.

In Line 386-387, authors state that “including more features would substantially increase model complexity without significantly affecting the conclusions of the study”, they should demonstrate that, in particular when considering virologic features (i.e., higher transmissibility/escape potential due to specific mutations).

Line 395: “confirmed cases are the currently available data for tracking the disease dynamics”, not sure since cases are definitely underestimated, they should give more weight to hospitalizations as a more reliable data in particular if the aim is to simulated general spatio-temporal disease dynamics.

In order to make clearer the message of the MS I would suggest to put more attention in reporting data. In particular, I found tricky the Figure 2, authors may want consider to show those data as table (?)

I would suggest to show the most interesting data in a better way in order to strengthen the bottom line of the message and the whole MS.

The y-axis in all the graphs (in a figure) may have the same scale in order to grab the difference if any.

I would really appreciate they discuss some of the limitations of the study, which would highlight a strong analytical sense and scientific responsibility, meaning that they got the weaknesses of the study and its design, I would suggest to rewrite the MS in order to get clear and more convincing narrative.

Author Response

In the manuscript "Implementation of a vaccination program based on epidemic geospatial
attributes: COVID-19 pandemic in Ohio as a case study and proof of concept” Susanne Awad
and colleagues determined which vaccine distribution strategy - on a spatial scale - would be the
most effective in minimizing both hospitalizations and new COVID-19 cases in Ohio (US) in
particular in a context characterized by limited resources (i.e., availability) during the early
stage of the vaccine rollout.

They use an innovative spatiotemporal model to assess the impact of vaccination distribution
strategies based on disease geospatial attributes and population-level risk assessment, and
demonstrated that a vaccine program that initially targets high infection intensity areas
compared to an equitable vaccination - has the most significant impact in reducing new cases
and infection-related hospitalizations.

Although the math under the study is beyond my competences I found very interesting and useful
this approach in particular in the context of a pandemic; however, some inaccuracies and gaps
particularly in the virologic facets are present and deserve attention.

Here some comments/concerns (among others).

Comment: We thank the reviewer for assessing our work and for the constructive feedback on
our manuscript that enriched the article and improved its readability. Please find below a point-
by-point reply addressing the reviewer’s comments.

1. The authors stated “this approach can be generalized to any other intervention strategy or
any region” therefore it could be very interesting and convincing if they show this approach
is valid in a different region (nation, why not) and also for other well-known diseases (i.e.
influenza, measles).

Answer: We apologize for the confusion. By this statement we meant that impact of intervention
distribution strategies based on geospatial attributes can be generalized to other disease and
countries. We agreed with the reviewer about the importance to show the applicability of this
approach to other regions and diseases. However, this work was an exploratory methodological

study, and thus we used COVID-19 as a study case and applying the methodology to another
diseases requires its own mathematical model structure based on its natural history of
transmission. To show this approach is valid for other well-known diseases is a separate study by
itself that is beyond the scope of this study. Therefore, we are currently working in other study
that applies this novel methodology to Cholera in Africa. Also, to address the reviewer’s concern
we have toned down this sentence and mentioned that further studies are needed to validate the
generalization of this approach. (lines 345-349 in the marked document).

2. Authors should consider the transmissibility variation in the context of the circulating
variants that represents a very important factor and make differences in the spreading of the
infection.

Answer: We want to highlight that we used COVID-19 as case of study to explore the
methodological approach discussed in this manuscript, but our intention was never to design
optimal vaccination programs for this disease. Therefore, several assumptions were made to
include the most relevant characteristics of the diseases to maintain the simplicity of the model.
Also we stated that data used for model fit come from the early stage of the epidemic, where
emergence of variants might not have substantial impact on the dynamics of the disease.
Assessment of the impact of the emergence of variants on the dynamics of the disease and
vaccination strategies is beyond the scope of or study. However, this would be an interesting
factor to study and thus we mentioned the importance of further studies exploring the emergence
of new variants in our revised manuscript (lines 167-173 and 421-424 in the marked document).

3. In Figure S1 (and also in the formula), authors should consider that deaths may occur also
out of ICU. Moreover, after dismission people may be susceptible and/or receive
vaccination. (Note: in the contest of high-transmissible variant, vaccination do not grant a
100% protection from infection this make the difference as variable).

Answer: Mortality out of ICU was included in the model (parameter
in Figure S1 and Table
S1. We assumed that infected individuals generated natural immunity after infection that lasted
during the time of the period evaluated (about 1 year). We have clarified this modeling
assumption in the revised version of the manuscript. (Lines 172-173 in the marked document).

4. Authors may define and explain what they refer to as susceptible.

Answer: SI (susceptible-infected) models like the one described in this study (which is an
expansion of an SI model) are well known and extensive used models since they were proposed
in 1917. Susceptible refers to individuals that can get the infection from an infected individual
and thus become infected. We have clarified this in the revised version of our manuscript (Lines
119-120 in the marked document).

5. Not sure about what is the bottom-line rationale of the Scenario 2.

Answer: Given that countries could make vaccine allocation decisions based on easy-to-reach
rather than on infection intensity, the impact of such choice is presented in scenario 2, and thus,
we wanted to compare the quantitative impact of such choice (Lines 199-201).

6. Authors should explain which is the weight of the containment measures, actually this is a
known and wide used index defined as stringency that they should include in the equation.

Answer: Containment measures were introduced as a time-variable infectivity rate in the model.
We want to highlight that this is a population-based mode and parameters estimated like
infectivity rated are population averages that capture several potential variations in the infection
rate produced by factors like containment measures.

7. In Line 367-369 authors stated, without reference, that “... for a successful vaccination
strategy, vaccines should be distributed equally in all areas and covering all population
groups”, not sure if it is sharp to state that in the context of a study where authors’ message
is different.

Answer: We apologize for the confusion. the statement has now been rephrased in the revised
Discussion (Lines 376-377 in the marked document).

8. In Line 386-387, authors state that “including more features would substantially increase
model complexity without significantly affecting the conclusions of the study”, they should
demonstrate that, in particular when considering virologic features (i.e., higher
transmissibility/escape potential due to specific mutations).

Answer: We referred to geospatial attributes rather than biological attributes, and we have
clarified this in the revised version of the manuscript (Line 396 in the marked document).
Assessment of the variation emerging from the virological attributes mentioned by the reviewer
are beyond the scope of our study and needs a different modeling approach that the one
discussed in our manuscript.

9. Line 395: “confirmed cases are the currently available data for tracking the disease
dynamics”, not sure since cases are definitely underestimated, they should give more weight
to hospitalizations as a more reliable data in particular if the aim is to simulated general
spatio-temporal disease dynamics.

Answer: As it was described in methods and results, we used data from not only cases but also
hospitalizations, ICU admissions and deaths to fit the model. To avoid confusion, we have
clarified this again in the paragraph mentioned by the reviewer (Lines 404-407 in the marked
document).

10. In order to make clearer the message of the MS I would suggest to put more attention in
reporting data. In particular, I found tricky the Figure 2, authors may want consider to show
those data as table (?)

Answer: We have followed the reviewer’s suggestion and included a table summarizing the
results illustrated in Figure 2 in Supplementary Information. (Table S3).

11. I would suggest to show the most interesting data in a better way in order to strengthen the
bottom line of the message and the whole MS.

Answer: We believe that the figures and maps included in the manuscript illustrate the most
important results from our study. However, as it was mentioned in the reply for the previous
comment, a table summarizing the results illustrated in Figure 2 in Supplementary Information
(Table S3).

12. The y-axis in all the graphs (in a figure) may have the same scale in order to grab the
difference if any.

Answer: We thank the reviewer for pointing this out. We have followed the reviewer’s
suggestion and use the same scale for Figure 1and Figure S3.

13. I would really appreciate they discuss some of the limitations of the study, which would
highlight a strong analytical sense and scientific responsibility, meaning that they got the
weaknesses of the study and its design, I would suggest to rewrite the MS in order to get
clear and more convincing narrative.

Answer: A subsection for limitations of the study was included in the revised version of the
manuscript. (Lines392-431).

Reviewer 4 Report

The research work carried out by the authors is very interesting and innovative. The research is well developed. The conclusions are missing. Data description and presentation is very confusing. There are grammatical errors throughout the manuscript. The rationale is not well explained. The discussion should be expanded to included a weaknesses paragraph. The introduction should be extended to make the manuscript suitable for publication.

Author Response

The research work carried out by the authors is very interesting and innovative. The research is
well developed. The conclusions are missing. Data description and presentation is very
confusing. There are grammatical errors throughout the manuscript. The rationale is not well
explained. The discussion should be expanded to included a weaknesses paragraph. The
introduction should be extended to make the manuscript suitable for publication.

Answer:
We thank the reviewer for the suggestions. We have now revised the manuscript to
address the reviewer’s concerns: Specifically, revised both the Introduction and the Discussion
section, and corrected any grammatical errors throughout (several instances throughout the
manuscript).

Round 2

Reviewer 3 Report

The authors satisfied several issues that I have highlighted in the first submission, and I found this revised MS version clearly improved. In fact, they mentioned and, in some cases, discussed properly several aspects of the limitations, resulting in a more solid MS.

Reviewer 4 Report

All issues have been addressed. No further suggestions or comments. The manuscript is acceptable for publication.